# Simplicity Level Estimate (SLE):
# A Learned Reference-Less Metric for Sentence Simplification

**Liam Cripwell**
Université de Lorraine
CNRS/LORIA
liam.cripwell@loria.fr

**Joël Legrand**
Université de Lorraine
Centrale Supélec
CNRS/LORIA
joel.legrand@inria.fr

**Claire Gardent**
CNRS/LORIA
Université de Lorraine
claire.gardent@loria.fr

## Abstract

Automatic evaluation for sentence simplification remains a challenging problem. Most popular evaluation metrics require multiple high-quality references – something not readily available for simplification – which makes it difficult to test performance on unseen domains. Furthermore, most existing metrics conflate simplicity with correlated attributes such as fluency or meaning preservation. We propose a new learned evaluation metric (SLE) which focuses on simplicity, outperforming almost all existing metrics in terms of correlation with human judgements.

## 1 Introduction

Text simplification involves the rewriting of a text to make it easier to read and understand by a wider audience, while still expressing the same core meaning. This has potential benefits for disadvantaged end-users (Gooding, 2022), while also showing promise as a preprocessing step for downstream NLP tasks (Miwa et al., 2010; Mishra et al., 2014; Štajner and Popovic, 2016; Niklaus et al., 2016). Although some recent work considers simplification of entire documents (Sun et al., 2021; Cripwell et al., 2023a,b) the majority of work focuses on individual sentences, given the lack of high-quality resources (Nisioi et al., 2017; Martin et al., 2020, 2021).

A major limitation in evaluating sentence simplification is that most popular metrics require high-quality references, which are rare and expensive to produce. This also makes it difficult to assess models on new domains where labeled data is unavailable. Another limitation is that many metrics evaluate simplification quality by combining multiple criteria (fluency, adequacy, simplicity) which makes it difficult to determine where exactly systems succeed and fail, as these criteria are often highly correlated — meaning that high scores could be spurious indications of simplicity (Scialom et al.,

| Metric | Simplification | Semantic | Ref-less |
|---|:---:|:---:|:---:|
| BLEU | ✗ | ✗ | ✗ |
| BERTScore | ✗ | ✓ | ✗ |
| QUESTEVAL | ✗ | ✓ | ✓ |
| SARI | ✓ | ✗ | ✗ |
| FKGL | ✓ | ✗ | ✓ |
| LENS | ✓ | ✓ | ✗ |
| SLE | ✓ | ✓ | ✓ |

Table 1: Desirable attributes of popular simplification evaluation metrics — whether they are designed with simplification in mind, use semantic representations, or do not require references.

2021b). Table 1 describes how popular metrics conform to various desirability standards.

We propose SLE (**S**implicity **L**evel **E**stimate), a learned reference-less metric that is trained to estimate the simplicity of a sentence.[1] Different from reference-based metrics (which estimate simplicity with respect to a reference), SLE can be used as an absolute measure of simplicity, a relative measure of simplicity gain compared to the input, or to measure error with respect to a target simplicity level. In this short paper, we focus on simplicity gain with respect to the input and show that SLE is highly correlated with human judgements of simplicity, competitive with the best performing reference-based metric. We also show that, when controlling for meaning preservation and fluency, many existing metrics used to assess simplifications do not correlate well with human ratings of simplicity.

## 2 A Metric for Simplicity

**The SLE Metric.** We propose SLE, a learned metric which predicts a real-valued simplicity level for a given sentence without the need for references. Given some sentence $t$, the system predicts a score $\text{SLE}(t) \in \mathbb{R}$, with high values indicating higher

---

[1] Code and resources are available at https://github.com/liamcripwell/sle/.

simplicity. This can not only be used as an absolute measure of simplicity for system output $\hat{y}$, but also to measure the simplicity gain relative to input $x$:

$$\Delta\text{SLE}(\hat{y}) = \text{SLE}(\hat{y}) - \text{SLE}(x) \quad (1)$$

In this paper we primarily focus on $\Delta\text{SLE}$, as it is the most applicable variant under common sentence simplification standards.

**Model.** As the basis for the metric, we fine-tune a pretrained RoBERTa model[2] to perform regression over simplicity levels given sentence inputs, using a batch size of 32 and $lr = 1e^{-5}$. We ran training experiments on a computing grid with a Nvidia A40 GPU.

**Data.** We use Newsela (Xu et al., 2015), which consists of 1,130 news articles manually rewritten at five discrete reading levels (0-4), each increasing in simplicity. Existing works often assume sentences have the same reading level as the document they are from (Lee and Vajjala, 2022; Yanamoto et al., 2022); however, we expect there to be a lot of variation in the simplicity of sentences within documents and overlap across levels. As such, merely training to minimize error with respect to these labels would likely result in mode collapse within levels (peaky, low-entropy distribution) and strong overfitting to the Newsela corpus. To address this mismatch between document- and sentence-level simplicity, we take the following two mitigating steps to allow the model to better differentiate between sentences from the same reading level.

**Label Softening.** We attempt to mitigate peakiness in the output distribution by softening the quantized reading levels assigned to each sentence in the training data. Specifically, we interpolate regression labels throughout overlapping class regions ($\pm 1$) according to their Flesch-Kincaid grade level (FKGL) (Kincaid et al., 1975). FKGL is a readability metric often used in education as a means to judge the suitability of books for students (high values $\implies$ high complexity).

If $L$ is the set of sentences belonging to some reading level, we define an intra-level ranking according to re-scaled, negative FKGLs:

$$f_L = \{-\text{fkgl}(x_i) \mid x_i \in L\}$$
$$f'_{L,i} = 2 \cdot \frac{f_{L,i} - \min f_L}{\max f_L - \min f_L} \quad (2)$$

where $f'_{L,i}$ is the revised FKGL score of sentence $x_i$. Intuitively, this inverts FKGL scores (so that higher values = higher simplicity) and rescales them to be $\in [0, 2]$. The $[0, 2]$ scaling is used in order for the distribution of final scores in each reading level to have a $\pm 1$ variance and overlap with adjacent groups (see Figure 1 for a visual representation).

From this, we derive the final revised labels:

$$l'_{L,i} = f'_{L,i} - \bar{f}'_L + l_{L,i} \quad (3)$$

where $\bar{f}'_L$ is the mean of $f'_L$, $l_{L,i}$ is the reading level for the $i$th sentence of $L$, and $l'_{L,i}$ is its revised soft version.[3]

For example, if the original document has a reading level of 3, and one of the sentences has a revised FKGL (Equation 2) of 1.5, then the softened label for that sentence will therefore be 3.5 (Equation 3). Figure 1 shows the distributional differences between the original reading levels and the resulting softened versions for the training data.

We report results for a model using softened labels (SLE) as well as a variant using the original quantized labels ($\text{SLE}_{\mathbb{Z}}$).

**Document-Level Optimization.** Given that Newsela reading levels are assigned at the document level, the labels of individual sentences are likely noisy, but approach the document label on average. We therefore observe and perform early stopping with respect to the document-level validation MAE (Mean Absolute Error) and use a train/dev/test split (90/5/5) that keeps sentences from all versions of a given article together. The size of each data split is illustrated in Table 2.

## 3 Experiments

### 3.1 Similarity Metrics

We compare SLE with four reference-based and two reference-less metrics previously used to assess the output of simplification models. Table 1 summarizes their main features.

**SARI.** The most commonly used evaluation metric is SARI (Xu et al., 2016), which compares $n$-gram edits between the output, input and references. Despite its widespread usage, SARI has known limitations. The small set of operations it considers

---

[2]We fine-tune the pretrained *roberta-base* model from https://huggingface.co/roberta-base with an added regression head.

[3]Note that we also account for extreme FKGL values by excluding sentences with FKGL scores in the top or bottom percentile.

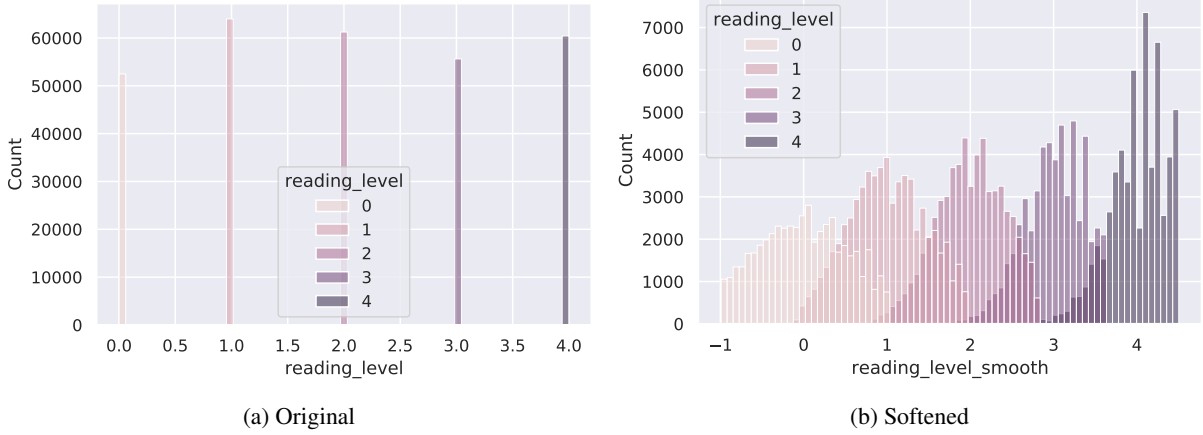

Figure 1: Distribution of (a) original quantized and (b) softened labels for sentences in the SLE training data.

|       | 0      | 1      | 2      | 3      | 4      | **Total** |
|-------|--------|--------|--------|--------|--------|-----------|
| Train | 52,488 | 63,993 | 61,248 | 55,620 | 60,416 | 293,765   |
| Dev   | 3069   | 3,595  | 3,348  | 2,976  | 3,228  | 16,216    |
| Test  | 2901   | 3,710  | 3,348  | 3,024  | 3,244  | 16,227    |
| **All** | **58,458** | **71,298** | **67,944** | **61,620** | **66,888** | **326,208** |

Table 2: Number of sentences sourced from documents of each quantized reading level.

makes it much more focused towards lexical simplifications, showing very low correlations with human ratings in cases where structural changes (e.g. sentence splitting) have occurred (Sulem et al., 2018). As it is token-based, it is totally reliant on the references, without any robustness towards synonomy.

**BERTScore.** Zhang et al. (2019) present BERTScore, which overcomes some of these shortcomings given its use of embeddings to compute similarities. It has been found to correlate highly with human ratings of simplicity (Alva-Manchego et al., 2021), but still requires references. It is reportedly worse than SARI at differentiating conservative edits (Maddela et al., 2022) and its high correlation with simplicity ratings may be spurious (Scialom et al., 2021b).

**LENS.** Recently, Maddela et al. (2022) propose LENS, a learnable metric specifically designed for simplification, which aims to better account for different operation types. It is trained to focus on semantic similarity without respect to writing style via a reference-adaptive loss. A simplification quality score is predicted given an output and a set of references. LENS shows higher correlations to human quality judgements than any previous metric, but still requires multiple references to work optimally.

**FKGL.** The Flesch-Kincaid grade level (FKGL) (Kincaid et al., 1975) is a document-level metric used to measure text readability without any references. It is based on basic surface-level features like word/sentence lengths. It has seen some success in evaluating simplification (Scialom et al., 2021b). Unlike most other metrics, it does not explicitly consider the adequacy and fluency dimensions, as it is reference-less and assumes the text is already well-formed (Xu et al., 2016).

**QUESTEVAL.** Scialom et al. (2021a) propose QUESTEVAL, a reference-less metric that compares two texts by generating and answering questions between them. Although originally intended for summarization, it has shown some promise as a potential meaning preservation metric for simplification (Scialom et al., 2021b).

### 3.2 Evaluation

We evaluate SLE both in terms of its ability to perform the regression task and how well it correlates with human judgements of simplicity. For the latter we consider $\Delta$SLE, as this conforms with what human evaluators were asked when giving ratings (to measure simplicity gain vs. the input).

**Regression.** To evaluate regression models we consider (i) the MAE with respect to the original quantized reading levels, (ii) the document-level error when averaging all sentence estimates from a given document (Doc-MAE), and (iii) the F1 score as if performing a classification task, after rounding estimates. We expect the best model for our purposes to achieve a lower Doc-MAE as it should better approximate true document-level simplicity labels in aggregate.

**Correlation with Human Simplicity Judgments.** We test the effectiveness of the metric by comparing its correlation with two datasets of human simplicity ratings: Simplicity-DA (Alva-Manchego et al., 2021) and Human-Likert (Scialom et al., 2021b). Simplicity-DA contains 600 system outputs, each with 15 ratings and 22 references, whereas Human-Likert contains 100 human-written sentence simplifications, each with ∼60 simplicity ratings and 10 references. We use all references when computing the reference-based metrics and consider the average human simplicity rating for each item.

As Simplicity-DA consists of system output simplifications, it naturally contains some sentences that are not fluent or semantically adequate. In such cases, humans would likely give low scores to the simplicity dimension as well (e.g. it is not simple to understand non-fluent text) — this is reflected in the inter-correlation between simplicity and the two other dimensions (Pearsons' $r$ of 0.771 for fluency and 0.758 for adequacy). Thus, we only consider a subset (Simplicity-DA✓) containing those system outputs with both human fluency and meaning preservation ratings at least 0.3 std. devs above the mean (top ∼30%)[4] which allows us to more appropriately consider how well metrics identify simplicity alone. For Human-Likert, the inter-correlation with fluency and meaning preservation are less pronounced, but do still exist (0.736 and 0.370).[5] As such, a metric with high correlation on Human-Likert but low correlation on Simplicity-DA✓ means it is likely measuring one of the other aspects rather than simplicity itself.

## 4 Results

Results on the regression task can be seen in Table 3. We see that although using soft labels ob-

| Model | MAE ↓ | Doc-MAE ↓ | F1 ↑ |
|---|---|---|---|
| SLE$_\mathbb{Z}$ (quantized) | 0.825 | 0.544 | 0.401 |
| SLE (softened) | 0.924 | 0.448 | 0.402 |

Table 3: Accuracy results for reading level estimators. Errors are calculated according to the original quantized reading level labels.

| Metric | Human-Likert | Simplicity-DA✓ |
|---|---|---|
| LENS | **0.531**\*\* | **0.429**\*\* |
| SARI | 0.395\*\* | 0.109 |
| BERTScore | 0.389\*\* | 0.142 |
| BLEU | 0.333\*\* | 0.084 |
| ΔSLE | **0.516**\*\* | **0.381**\*\* |
| ΔSLE$_\mathbb{Z}$ | 0.479\*\* | 0.328\*\* |
| FKGL | 0.354\*\* | 0.260\* |
| QUESTEVAL | 0.134 | 0.090 |

Table 4: Absolute Pearson correlations with human judgements of simplicity. The top tier contains reference-based metrics and the bottom reference-less. * indicates significance with $p$-value < 0.01 and ** < 0.001.

viously worsens MAE with respect to the original reading levels, the document-level MAE is improved, suggesting that quantized labels lead to more extreme false negatives under uncertainty, as scores are drawn towards integer values. When treated as a classification task (by rounding predictions) both systems show similar performance (F1). This shows us that SLE is better able to approximate document-level simplicity ratings on average, with little to no drawback at the sentence level (assuming quantized labels were correct).

Correlations with human ratings of simplicity are shown in Table 4. The best metric on Human-Likert is LENS, closely followed by ΔSLE, with other metrics lagging quite far behind. This clearly shows the effectiveness of ΔSLE as it is able to outperform all existing metrics but for LENS, without requiring any references and using a smaller network architecture than LENS and BERTScore. On Simplicity-DA✓, metrics follow a similar rank order except for certain metrics dropping substantially (SARI, BERTScore, BLEU).[6] As Human-Likert still has moderate inter-correlation between evaluation dimensions, the large drops in performance can likely be attributed to these mostly mea-

---

[4]We also exclude 166 examples that exist within the LENS training data.

[5]We do not perform any filtering on Human-Likert.

[6]Scialom et al. (2021b) report very poor reference-based metric correlations on Human-Likert, substantially lower than our results. When discussed with the authors, they were no longer in possession of code that could reproduce their originally reported results.

suring semantic similarity with references rather than the actual simplicity. Accounting for the inter-correlation between dimensions has less impact on metrics like $\Delta$SLE and FKGL, confirming the validity of readability-based metrics as potential measures of pure simplicity.

## 5 Related Work

Štajner et al. (2014) attempt to assess each quality dimension of simplifications by training classifiers of two (good, bad) or three (good, medium, bad) classes using existing evaluation metrics as features. However, when the simplicity dimension is considered, performance was poor (Štajner et al., 2016). Later, Martin et al. (2018) were able to slightly improve this after exploring a wide range of features. However, these works do not predict real-valued estimates of simplicity nor have been adopted as evaluation metrics.

Some studies from the automatic readability assessment (ARA) literature use quantized Newsela reading levels as labels to train regression models. Lee and Vajjala (2022) do so in order to predict the readability of full documents, which does not extend to sentence simplification. Yanamoto et al. (2022) predict a reading level accuracy within an RL reward for sentence simplification, but do so using the reading levels that were assigned to each document. This too does not transfer well to sentence-level evaluation, given the imprecision and noise introduced by the use of quantized ratings that were assigned at the document level. These approaches have not been applied to the actual evaluation of sentence simplification systems.

## 6 Future Directions

In this paper we explore the efficacy of SLE as a measure of raw simplicity or relative simplicity gain ($\Delta$SLE). However, given the flexibility of not relying on references, SLE can potentially be used in other ways. For example, one could measure an error with respect to a target simplicity level, $l^*$:

$$\epsilon\text{SLE}(\hat{y}) = |\text{SLE}(\hat{y}) - l^*| \tag{4}$$

This could be useful in the evaluation of controllable simplification systems, which should be able to satisfy simplification requirements of specific user groups or reading levels (Martin et al., 2020; Cripwell et al., 2022; Yanamoto et al., 2022). As it is trained with aggregate document-level accuracy

in mind, SLE could also possibly be used to evaluate document simplification — either via averaging sentence scores or using some other aggregation method.

## 7 Conclusion

In this paper we presented SLE — a reference-less evaluation metric for sentence simplification that is competitive with or better than the best performing reference-based metrics in terms of correlation to human judgements of simplicity. We reconsider the ability of popular metrics to accurately gauge simplicity when controlling for other factors such as fluency and semantic adequacy, confirming suspicions that many do not measure simplicity directly. We hope this work motivates further investigation into the efficacy of standard simplification evaluation techniques and the proposal of new methodologies.

## 8 Acknowledgements

We thank the anonymous reviewers for their feedback. We gratefully acknowledge the support of the French National Research Agency (Gardent; award ANR-20-CHIA-0003, XNLG "Multilingual, Multi-Source Text Generation").

Experiments presented in this paper were carried out using the Grid'5000 testbed, supported by a scientific interest group hosted by Inria and including CNRS, RENATER and several Universities as well as other organizations (see `https://www.grid5000.fr`).

## 9 Limitations

The SLE metric model is trained entirely on English-language data and therefore will not be effective for evaluating simplification in other languages. Producing a multilingual version of the metric is likely possible by using either different datasets or adapting other methods from the multilingual NLP literature, but we leave this to future work.

Furthermore, as SLE has been primarily trained on news articles, it could exhibit a drop in performance quality when used to evaluate text from specialized domains that use vocabularies likely not encountered during training (e.g. medical, legal domains). In such cases, producing a domain-specific version of SLE via specialized pretraining or fine-tuning should be feasible, given sufficient data.

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

| Model | MAE ↓ | Doc-MAE ↓ | F1 ↑ |
|---|---|---|---|
| SLE$_{\mathbb{Z}}$ (quantized) | 0.820 | 0.543 | 0.404 |
| SLE (softened) | 0.931 | 0.457 | 0.405 |

Table 5: Accuracy results for reading level estimators on the validation set. Errors are calculated according to the original quantized reading level labels.

Sanja Štajner, Maja Popovic, Horacio Saggion, Lucia Specia, and Mark Fishel. 2016. Shared task on quality assessment for text simplification. *Training*, 218(95):192.

Elior Sulem, Omri Abend, and Ari Rappoport. 2018. Semantic structural evaluation for text simplification. In *Proceedings of the 2018 Conference of the North American Chapter of the Association for Computational Linguistics: Human Language Technologies, Volume 1 (Long Papers)*, pages 685–696, New Orleans, Louisiana. Association for Computational Linguistics.

Renliang Sun, Hanqi Jin, and Xiaojun Wan. 2021. Document-level text simplification: Dataset, criteria and baseline. In *Proceedings of the 2021 Conference on Empirical Methods in Natural Language Processing*, pages 7997–8013, Online and Punta Cana, Dominican Republic. Association for Computational Linguistics.

Wei Xu, Chris Callison-Burch, and Courtney Napoles. 2015. Problems in current text simplification research: New data can help. *Transactions of the Association for Computational Linguistics*, 3:283–297.

Wei Xu, Courtney Napoles, Ellie Pavlick, Quanze Chen, and Chris Callison-Burch. 2016. Optimizing statistical machine translation for text simplification. *Transactions of the Association for Computational Linguistics*, 4:401–415.

Daiki Yanamoto, Tomoki Ikawa, Tomoyuki Kajiwara, Takashi Ninomiya, Satoru Uchida, and Yuki Arase. 2022. Controllable text simplification with deep reinforcement learning. In *Proceedings of the 2nd Conference of the Asia-Pacific Chapter of the Association for Computational Linguistics and the 12th International Joint Conference on Natural Language Processing (Volume 2: Short Papers)*, pages 398–404, Online only. Association for Computational Linguistics.

Tianyi Zhang, Varsha Kishore, Felix Wu, Kilian Q. Weinberger, and Yoav Artzi. 2019. Bertscore: Evaluating text generation with BERT. *CoRR*, abs/1904.09675.

## A    Validation Set Results

Table 5 lists the model performances on the validation set. Notice they follow a very similar pattern to what is seen in the test set results in Table 3.

## B    Regression Entropy

Figure 2 shows the distributional difference between the predictions from the SLE model with softened vs. quantized labels. Using the softened labels leads to predictions following a much smoother distribution.

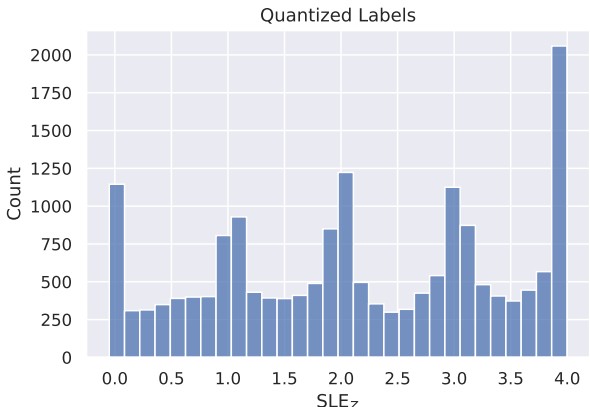

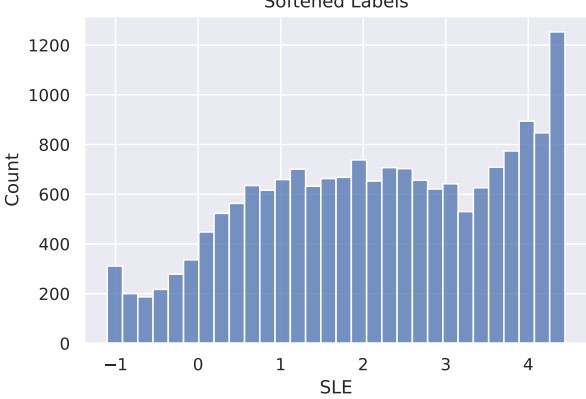

Figure 2: Distribution of test set predictions from SLE models trained on quantized vs. softened labels.