# OpenReview forum: "Simplicity Level Estimate (SLE): A Learned Reference-Less Metric for Sentence Simpliﬁcation"
_EMNLP/2023/Conference — EMNLP 2023 Main_

### Official Review · Reviewer_1Hnr · 2023-07-30

**Soundness:** 3

**Excitement:**

4: Strong: This paper deepens the understanding of some phenomenon or lowers the barriers to an existing research direction.

**Paper Topic And Main Contributions:**

The authors propose Simplicity Level Estimate (SLE), a referenceless metric for sentence simplification. SLE is finetuned on a pretrained RoBERTa model for regression on simplicity levels given sentence inputs on Newsela. Label softening is used to soften the reading levels assigned to each sentence, and document-level optimization is performed by early stop with respect to the document-level validation MSE. The metric considers both simplification and semantics, outperforms all other referenceless metrics, and is competitive with state-of-the-arts reference-based metrics.

**Questions For The Authors:**

A. Can SLE generalize to out-of-distribution data? As it is trained on only a thousand news articles, would it be hard to generalize to more oral text like twitter texts?

**Reasons To Accept:**

1. The real valued simplicity measure SLE is intuitive and performs very well as a referenceless measure, even competitive with reference based measures (outperforming all except LENS).
2. The proposed metric SLE is less affected when accounting for inter-correlation between dimensions. The authors have good analysis on cases when simplicity dimensions are correlated.

**Reasons To Reject:**

1. Unsure how well SLE generalizes if trained on news data.

**Reproducibility:**

4: Could mostly reproduce the results, but there may be some variation because of sample variance or minor variations in their interpretation of the protocol or method.

**Reviewer Confidence:**

3: Pretty sure, but there's a chance I missed something. Although I have a good feel for this area in general, I did not carefully check the paper's details, e.g., the math, experimental design, or novelty.

---

> ### Author Rebuttal · Authors · 2023-08-28
>
> Thank you very much for taking the time to review our paper.
>
> The datasets we evaluate the metrics on (Human-Likert and SimplicityDA) are actually not from news data, so the evaluation in the paper does capture the generalizability of SLE. Both of those datasets use simplifications of sentences from the ASSET and TurkCorpus datasets, both of which are sourced from Wikipedia articles. This could actually be considered as another potential strength of SLE over other learned metrics like LENS (which was trained on the same Wikipedia domain we evaluate on).
>
> As you suggest, if the domain shifts further towards twitter comments or transcribed speech, it is likely that the performance will diminish, but this would probably be true for other learned metrics as well. However, we will make our training code publicly available so that newer versions could be trained on wider or more specialized data in the future.
>
> We hope that this clarifies the concern you raised in your reason to reject. Thank you once again for taking the time to review our work.

---

### Official Review · Reviewer_eCtA · 2023-08-05

**Soundness:** 4

**Excitement:**

4: Strong: This paper deepens the understanding of some phenomenon or lowers the barriers to an existing research direction.

**Paper Topic And Main Contributions:**

Evaluation metric for Sentence Simplication task. Named SLE, is a model based (i.e. learned) metric. Finetune Roberta on a datatset (Newsela) that has 1130 new articles rewritten to 5 reading levels. There are some challenges,e.g., rewritten docs have mixture of levels among sentences. Authors propose a method to tackle this challenge: label softening and doc-level optimization.

Contributions:
*  They develop a (learned) metric for simplicity evaluation (SLE). They describe methods to tackle the challenge of mismatch in sentence-level and document-level simplicity annotation. _Note: proposed SLE metric is reference less._
* They conduct (meta) evaluation. The results show that SLE is the best reference-free metric, a runner up only to the reference based LENS metric. It has stronger correlations with human scores than many other commonly used reference based metrics such as SARI, BERTScore etc.


**Questions For The Authors:**

#Q1) Equation 2. Why do we need scalar multiplier 2 that results in rescaled value of [0,2]. Why not [0, 1] and hence no scalar multiplier (Occam’s razar)?

#Q2)Seems like roberta-base model is used. Did you try roberta-large as pretrained model? Considering that GPUs are getting faster over years, large maybe the new base, and some users may be able to afford extra cost for higher performance.


**Reasons To Accept:**

The proposed metric appears to have strong correlations.  Paper is easy to read and follow. The approach is simple and elegant. See Contributions in the above paragraph (which seems sufficient for a short paper). Assuming authors would provide finetuned models for use, it would be a useful addition to community.

**Reasons To Reject:**

There seem no major flaws in methodology and analysis. However, a nitpick; in Table 3, results are based on pearson correlation, which is known to suffer from outliers in the sample set. While the authors give ranking on two datasets, nearly half of metrics don't have a statistically significant correlation on Simplicity-DA, so we not thoroughly convinced about the usefulness of the dataset. Having Kendal’s Tau in addition to pearson r would be one way of boosted our confidence in the results, or another dataset where most commonly used nmetrics have statistically sound correlations.

**Reproducibility:**

5: Could easily reproduce the results.

**Reviewer Confidence:**

1: Not my area, or paper was hard for me to understand. My evaluation is just an educated guess.

**Typos Grammar Style And Presentation Improvements:**


Figures in Page 7 maybe possible to move into the whitespace in Page6. (Option 1: subfigures one above the other. Option 2: `\usepackage{balance}`)

---

> ### Author Rebuttal · Authors · 2023-08-28
>
> Thank you very much for taking the time to review our work.
>
> Regarding your point on the low correlation for some of the metrics on SimplicityDA: we view this as a being a result of those metrics (SARI, BertScore, Bleu, QuestEval) not actually measuring simplicity, but only the inter-correlated attributes (e.g. meaning preservation) that we are removed from the SimplicityDA data (cf Sec. 3.2) but which still exist in the  Human-Likert dataset as the correlation coefficients for those metrics suddenly increase a lot on the Human-Likert data. In contrast, the fact that Pearson and significance remains high on both datasets for these metrics that are more specifically focused on simplicity (SLE, LENS, FKGL) gives us confidence that these are quite reliable metrics as they have good correlation on both datasets.
>
> Question 1.
>
> Using an interval of [0,1] would mean scores are softened by a +-0.5 factor. This would yield a much peakier final distribution that would have very little representation for values on the borders before reading levels (see Figure 1 for visual context). We use the [0,2] scaling so that the distribution of scores in each reading level has a wider variance and overlap with adjacent groups (+-1.0).
>
> Question 2.
>
> Yes, we initially considered using roberta-large (this is what LENS and BERTScore are based on) but switched to roberta-base due to us having  restrictive GPU access. Training larger versions of the model would definitely be something we’d encourage or even try to do ourselves in the future and we will be releasing all of our training code to allow for this.

---

### Official Review · Reviewer_SwXe · 2023-08-08

**Soundness:** 4

**Excitement:**

4: Strong: This paper deepens the understanding of some phenomenon or lowers the barriers to an existing research direction.

**Paper Topic And Main Contributions:**

This paper presents a new evaluation measure for simplicity in the context of sentence simplification evaluation. The measure is learnable, using the Newsela labels together with label softening as a basis for training a RoBERTa-based regression model and computes the simplicity relative to the input (but can also address absolute simplicity) without using references.
The new measure called SLE achieves the highest correlation with human evaluation on two benchmarks, among the reference-less measures. The results also show the difficulty of existing measures that achieve high overall correlations to capture the simplicity dimension of the task.

**Questions For The Authors:**

In continuation with the discussion on SARI (lines 135-146), to what extent does SLE capture structural simplicity? Or, in other words, to what extent is structural simplicity present in the Newsela dataset on which SLE learns and on the Simplicity-DA and Human-Likert Benchmarks on which it is evaluated?

**Reasons To Accept:**

- The paper is very well-written and well-presented.
- It represents an interesting contribution to sentence simplification evaluation, following the ideas of the use of a separate measure for the simplicity dimension of sentence simplification and of the ability to evaluate simplicity without using references.
- The connection between the sentence-level and document-level simplicity as done in the computation of the measure seems very useful.


**Reasons To Reject:**

A drawback of the approach is that it is opaque in the sense that the measure does not tell us in which way the output is simpler than the input, for example in terms of operations performed or in terms of lexical vs. structural simplicity. I think it would be useful to discuss it in the paper.

**Reproducibility:**

4: Could mostly reproduce the results, but there may be some variation because of sample variance or minor variations in their interpretation of the protocol or method.

**Reviewer Confidence:**

4: Quite sure. I tried to check the important points carefully. It's unlikely, though conceivable, that I missed something that should affect my ratings.

**Typos Grammar Style And Presentation Improvements:**

Presentation:

- It would be useful to present example of outputs that get high scores and low scores by SLE, also comparing with other metrics.
- I think the differences between LENS and SLE, beyond the use of references, should be described in more details.
- It would be interesting to also report the scores for the references.
- It would be useful to extend the part on future directions (the adaptation of the measure to a target simplicity level) to a discussion about reference-less measures, also mentioning a recent work on the limitations of reference-less measures for text generation:
"On the Limitations of Reference-Free Evaluations of Generated Text", Daniel Deutsch, Rotem Dror and Dan Roth, Proceedings of EACL 2023, pages 10960 - 10977.

Minor:

- Footnote 4: "the" should be removed

- line 282: it would be useful to explicitly write Reinforcement Learning.

---

> ### Author Rebuttal · Authors · 2023-08-28
>
> Thank you very much for taking the time and effort to review our paper
>
> Yes, you raise a good point re interpretability of e.g. structural vs lexical simplicity. As it stands SLE doesn’t really provide any clear indication of this as it is just a single score. However, we are fairly confident that it has learned to consider both of these factors as Newsela contains quite a diverse range of transformations (~24% of sentences are split). Also the data evaluated on does contain instances that include sentence splitting as they are derived from ASSET, which was specifically built to cover a wide array of possible transformation types. In SimplicityDA, 25% of references contain a sentence split and 73% of items have at least one reference with a split, however, splits are only in ~7.3% of outputs. In Human-Likert, there are a lot more with ~18% outputs being splits.
>
> Re presentation improvements: including example outputs and scores for references is a good idea that we will aim to take into account for the final version. Using some of the extra space to extend discussion of future directions would also be valuable.
>
> Thank you again for taking the time to review our paper.

---

### Official Review · Reviewer_bXBN · 2023-08-10

**Soundness:** 4

**Excitement:**

3: Ambivalent: It has merits (e.g., it reports state-of-the-art results, the idea is nice), but there are key weaknesses (e.g., it describes incremental work), and it can significantly benefit from another round of revision. However, I won't object to accepting it if my co-reviewers champion it.

**Missing References:**

There are no missing references. Avoid citing published works from arXiv: Martin et. al., 2021 is cited from arXiv, the LREC version should be cited instead.

**Paper Topic And Main Contributions:**

This paper proposes SLE, an automatic metric for evaluating sentence simplification in a reference-less manner. It argues that existing sentence simplification metrics combine various criteria instead of focusing on simplicity, and describes how SLE does this by design while also providing a way to measure simplicity gain. The key ideas are: (i) regressing over simplicity levels, (ii) obtaining soft labels by mixing simplicity levels with Flesch-Kincaid readability scores, and (iii) document-level optimization due to noisy sentence simplicity labels. The experiments show that SLE better correlates with human evaluations than existing metrics, and the authors explain that it will be publicly released once the paper is published.

**Questions For The Authors:**

Question A: Why do you multiply by 2 in the FKGL scores transformation to reach the [0, 2] range instead of [0, 1]? I infer this is related to the subtraction of the mean to approximate the range of simplicity level labels, but it is unclear.

Question B: What is the motivation behind the sum of transformed FKGL scores and simplicity levels in eq. (3)? Some brief insights would help in understanding why SLE correlates well with human evaluations.

Question C: Do you plan to release the code and data in addition to the metric itself?

**Reasons To Accept:**

The paper introduces a new state-of-the-art metric sentence simplicity metric that better correlates with human evaluations, using a novel technique to leverage document-level annotations for attributing sentence-level simplicity. The derivations are sound, and the experiments show that the novel smoothing technique allows for better performance than regressing over discrete levels. This technique could be further generalized to other tasks with per-level annotations. Additionally, the experiments support the claim that SLE correlates more closely with human evaluations than other methods. The paper is very well written, easy to read, and the motivation, arguments, experiments, and conclusions are communicated effectively.

**Reasons To Reject:**

The specific solutions to some problems lack technical motivation. Examples include: (i) scaling FKGL scores to [0, 2], and (ii) summing FKGL scores and simplicity levels. This makes it difficult to understand whether the novel ideas were obtained by design or discovered through trial.

Regarding the experimentation, the authors argue that one of the evaluation datasets for human correlations, Simplicity-DA, includes undesirable qualities. These are removed for evaluation, but it would have been interesting to see SLE’s (and the other metrics’) performance including sentences with these qualities to study its robustness.

Hardware set-up, training parameters, and validation set results are not reported: the paper may lack sufficient information to effectively reproduce the reported results, especially since one of the novel ideas includes early stopping with the validation set results.

**Reproducibility:**

4: Could mostly reproduce the results, but there may be some variation because of sample variance or minor variations in their interpretation of the protocol or method.

**Reviewer Confidence:**

4: Quite sure. I tried to check the important points carefully. It's unlikely, though conceivable, that I missed something that should affect my ratings.

**Typos Grammar Style And Presentation Improvements:**

Throughout the paper: remove spaces before footnotes. Additionally, “referenceless” can be used, without hyphen.

The notation can be made more consistent. Consider changing “t” to “x” in line 064, and ensure that the output co-domain is correctly defined in line 065 to avoid confusion, otherwise one might understand SLE to output scores in (-inf, inf), which I understood to not be the case given the existing sentence simplicity levels it was trained on.

Furthermore, line 104 defines L as the set of sentences belonging to some reading level, but then lowercase L is used in eq (3) to denote a sentence’s level. I would opt for a different letter, e.g. S, for the set of sentences, and introduce the lowercase L as a restriction in eq (2) when defining the set f_L. Overall, the Label Softening section was difficult to parse. The paper would greatly benefit from a better presentation of its technical aspects.

---

> ### Author Rebuttal · Authors · 2023-08-28
>
> Thank you very much for taking the time to review our paper.
>
> Re reasons to reject:
>
> 1.
>
> The [0,2] interval is used so that there is an approximate +-1 rescaling (see lns 97-100) around the original document simplicity level (Equation 3) and this was chosen by design (we want the different reading level groups to overlap with each other). In Equation 3 you can see that this is subtracted by the mean, $\bar{f^\prime_L}$ (~1), which then obviously yields the +-1 distribution. These values are then summed with the original simplicity level (0-4) to have them rescaled to be +-1 of that value. E.g. if the original document has a reading level of 3, and one of the sentences has a revised FKGL (Equation 2) of 1.5, then the softened label for that sentence will therefore be 3.5 (i.e., 1.5 - 1 + 3). Fig 1 (App, p7) illustrates this visually by showing the distributions of the softened labels.
>
> 2.
>
> The correlations of all metrics (except SLE) for the full SimplicityDA dataset can be seen in the original SimplicityDA paper. We feel like trying to include this in our short paper would be redundant and take up an unnecessary amount of space, especially considering our point is that these results are flawed anyway. SLE unsurprisingly has very low correlations there because it is impossible for it to consider meaning preservation (it doesn’t see references or inputs) and is only trained with fluent human-written data, so obviously isn’t equipped to determine disfluency either. See lines 214-230 for our rationale behind this move (also the thorough analysis in Scialom et al, 2021b).
>
> We feel like this point about robustness doesn’t really apply as the inclusion of those results would mean we would effectively be evaluating SLE on tasks that it is not trained to perform (meaning preservation and fluency) and also architecturally unable to handle the inputs that would be necessary to do so (meaning preservation). We also include results on another dataset (Human-Likert) which contains higher quality texts while still having some inter-correlation between the dimensions, so this should provide a good indication of robustness without using completely unsuitable data.
>
> 3.
>
> We do mention our training parameters in the paper (lines 73-76). As for the hardware setup, we didn’t mention this in this version of the paper as we thought it could lead to anonymity issues, but it was done using a fairly modest GPU. We will also be releasing our full training code so reproducibility shouldn’t be a problem (the training setup is not much more complicated than a standard regression model).
>
> The scores on the validation data are listed below. As you can see, they follow a very similar pattern to the test set results.
>
> ```bash
> Model   MAE    Doc-MAE  F1
> SLE_Z   0.820  0.543    0.404
> SLE     0.931  0.457    0.405
> ```
>
> Question A:
>
> This is more or less answered above in point 1. Using an interval of [0,1] would mean scores are softened by a +-0.5 factor. This would yield a much peakier final distribution that would have very little representation for values on the borders between reading levels (see Figure 1 for visual context). We use the [0,2] scaling so that the distribution of scores in each reading level has a wider variance and overlap with adjacent groups (+-1.0).
>
> Question B: See response to point 1 above.
>
> Question C: Yes, we plan to release the full code, model, and data (and data generation code). An anonymized version of much of this is also already included in the supplementary materials, but a polished and enhanced version will be put on Github by the time of publication.
>
> Re typos/presentation and missing references: Thanks for pointing those out, we will make sure to address them in the final version of the manuscript.
>
> Thank you again for your review of our paper and we hope we have sufficiently clarified any concerns you may have had.

---

### Meta-Review · Area_Chair_tpfK · 2023-09-16

**Recommendation:** 4

**Metareview:**

The authors propose a new learned evaluation metric for sentence simplification, SLE, which focuses on simplicity and outperforms almost all existing metrics in terms of correlation with human judgments.

---

### Decision · Program_Chairs · 2023-10-07

**Decision:**

Accept-Main

**Comment:**

The authors propose a new learned evaluation metric for sentence simplification, SLE, which focuses on simplicity and outperforms almost all existing metrics in terms of correlation with human judgments.